# Ursolic Acid against Prostate and Urogenital Cancers: A Review of In Vitro and In Vivo Studies

**DOI:** 10.3390/ijms24087414

**Published:** 2023-04-18

**Authors:** Amanda Kornel, Matteo Nadile, Maria Ilektra Retsidou, Minas Sakellakis, Katerina Gioti, Apostolos Beloukas, Newman Siu Kwan Sze, Panagiota Klentrou, Evangelia Tsiani

**Affiliations:** 1Department of Health Sciences, Faculty of Applied Health Sciences, Brock University, St. Catharines, ON L2S 3A1, Canada; 2Department of Kinesiology, Faculty of Applied Health Sciences, Brock University, St. Catharines, ON L2S 3A1, Canada; 3Department of Medical Oncology, Metropolitan Hospital, 18547 Athens, Greece; 4Department of Biomedical Sciences, School of Health Sciences, University of West Attica, 12243 Athens, Greece; 5National AIDS Reference Centre of Southern Greece, School of Public Health, University of West Attica, 11521 Athens, Greece; 6Centre for Bone and Muscle Health, Applied Health Sciences, Brock University, St. Catharines, ON L2S 3A1, Canada

**Keywords:** ursolic acid, prostate, urogenital, cancer, survival, apoptosis, proliferation, signaling cascades

## Abstract

Prostate cancer is the second most diagnosed form of cancer in men worldwide and accounted for roughly 1.3 million cases and 359,000 deaths globally in 2018, despite all the available treatment strategies including surgery, radiotherapy, and chemotherapy. Finding novel approaches to prevent and treat prostate and other urogenital cancers effectively is of major importance. Chemicals derived from plants, such as docetaxel and paclitaxel, have been used in cancer treatment, and in recent years, research interest has focused on finding other plant-derived chemicals that can be used in the fight against cancer. Ursolic acid, found in high concentrations in cranberries, is a pentacyclic triterpenoid compound demonstrated to have anti-inflammatory, antioxidant, and anticancer properties. In the present review, we summarize the research studies examining the effects of ursolic acid and its derivatives against prostate and other urogenital cancers. Collectively, the existing data indicate that ursolic acid inhibits human prostate, renal, bladder, and testicular cancer cell proliferation and induces apoptosis. A limited number of studies have shown significant reduction in tumor volume in animals xenografted with human prostate cancer cells and treated with ursolic acid. More animal studies and human clinical studies are required to examine the potential of ursolic acid to inhibit prostate and other urogenital cancers in vivo.

## 1. Introduction

Prostate cancer is the second most diagnosed form of cancer in men worldwide (second to lung cancer). It is responsible for nearly 400,000 deaths annually, approximately 4% of cancer related deaths in men [1]. Prostate cancer is associated with age with the highest incidence seen in men over 65 years of age; other contributing factors include diet, obesity, genetic/family history, and history of sexually transmitted disease. Early stages are asymptomatic, making diagnosis difficult. Later stages are associated with increased frequency and difficulty of urination, nocturia, and back pain [1]. Detection is based on elevated plasma levels of prostate-specific antigen (PSA) being greater that 4 ng/mL and confirmed with tissue biopsy [1]. The five year survival rate is approximately 98% in the USA, 76% in eastern Europe, and 88% in south and central Europe [1].

Prostate cancer is classified as either androgen sensitive or androgen insensitive. Androgens such as testosterone promote growth of the prostate gland. At the cellular level, the binding of androgens to androgen receptors (AR) leads to nuclear translocation of the AR–androgen complex which acts as a transcription factor activating genes promoting cell proliferation as well as the synthesis of prostate-specific antigen (PSA) [2]. Early stage prostate cancer is driven by AR activation and therefore androgen deprivation therapy is an effective treatment strategy at this stage [3]. However, in late-stage prostate cancer and following androgen deprivation therapy, cancer cells adapt to low AR–androgen complex signaling and activate alternative signaling cascades leading to survival, proliferation, and metastasis [3]. Typically, later-stage prostate tumors are androgen-insensitive and represent a more aggressive disease phenotype.

The phosphatidylinositol 3-kinase (PI3K)/Akt pathway plays a significant role in prostate cancer. Expression/activation of Akt is often elevated [4,5] and aberrations in this pathway have been reported in approximately 70–100% of advanced cases of the disease [6,7]. Activation of PI3K/Akt leads to activation of the mechanistic target of rapamycin (mTOR) and downstream promotion of protein synthesis and cell proliferation. Prostate cancer tissues were found to have overactivated mTOR when compared to normal prostate epithelium [8]. Mutations that result in activation of other oncogenes and/or the inactivation of proteins that serve as tumor suppressors such as p53, p27, and phosphatase and tensin homologue (PTEN) [9,10] also contribute to the development of prostate cancer [11,12,13].

Another characteristic of cancer cells, including prostate cancer cells, is the suppression of the process of programmed cell death known as apoptosis. Activation of both the intrinsic and extrinsic apoptotic pathways lead to the cleavage/activation of caspases and downstream cleavage of poly (ADP-ribose) polymerase-1 (PARP-1) and induction of apoptosis [14,15]. The development of agents that induce apoptosis in cancer cells is an active area of research and will result in better outcomes for cancer treatments including prostate cancer.

Apart from androgen deprivation therapy, other treatment strategies for prostate cancer include surgery/prostatectomy, radiation therapy, and chemotherapy. The treatment approach for each patient depends on the stage of the disease. In advanced in situ carcinoma cases, surgery or radiotherapy are the preferred treatment options. Although surgery reduces the potential for metastasis, it is unfortunately not associated with a high reduction in the mortality risk after 10 years. Radiotherapy, specifically external-beam radiotherapy and brachytherapy, are effective in treating localized tumors and are associated with higher life expectancy/lower mortality risk after 10 years [16,17]. All currently available treatment approaches are associated with adverse effects. Androgen deprivation therapy for example is associated with erectile dysfunction, hot flashes, anemia, and depression [18]. Although chemotherapy has been shown to be effective in patients with androgen-insensitive tumors, it is associated with diarrhea, nausea/vomiting, loss of appetite, and fatigue.

Many drugs used in chemotherapy are derived from plants. For example, the chemotherapy drugs paclitaxel and docetaxel, used in prostate cancer treatment, were originally isolated from the bark of the Pacific yew tree (*Taxus brevifolia*) and the needles of the European yew tree (*Taxus baccata*), respectively [19,20].

Finding novel approaches to prevent and treat prostate and other urogenital cancers effectively is highly desirable and the search for plant-derived chemicals with a strong anticancer potential is ongoing and hopefully will result in novel agents that will overcome the resistance to current chemotherapy drugs [21]. Plant-derived chemicals such as polyphenols found in tea and wine may target molecules such as sphingosine-1 phosphate [22], an important player in cancer progression and metastasis, resulting in effects against prostate cancer [23]. In addition, other plant-derived bioactive compounds such as berberine and matrine, may affect microRNA expression resulting in inhibition of tumor growth [24].

Ursolic acid (UA) is found in the leaves and fruits of many plant species with high concentrations detected in lavender, marigold, rosinweed, basil, rosemary, and olive tree leaves. In addition, fruits such as cranberries, black elderberries, apples, and pears [25] contain substantial levels of UA [26,27,28,29].

The chemical formula of UA is C_30_H_48_O_3_, and structurally, UA is classified as a pentacyclic triterpenoid (Figure 1). Evidence indicates that UA exhibits anti-inflammatory [30,31], neuroprotective [32,33], antidiabetic [34], and anticancer properties [8,35,36,37].

In this review article, we have summarized studies that examined the effects of UA against prostate and other urogenital (or genitourinary) cancers. A search was performed in PUBMED.com for the terms “ursolic acid and prostate cancer”, “ursolic acid and renal cancer”, “ursolic acid and bladder cancer”, and “ursolic acid and urogenital cancers”. Articles which specifically examined UA and these cancers were included and presented in chronological order.

## 2. Effects of Ursolic Acid against Prostate Cancer

### 2.1. Effects of Ursolic Acid against Prostate Cancer: Evidence from In Vitro Studies

The treatment of human androgen-sensitive LNCaP and hormone refractory PC-3 prostate cancer cells with UA (55 µM for 24 and 48 h) reduced viability (MTT assay) and induced apoptosis (Annexin assay) and these effects were associated with a downregulation of the anti-apoptotic protein B-cell lymphoma 2 (Bcl-2) [38] (Table 1). These results provide evidence that UA initiates apoptosis in prostate cancer cells through downregulating the anti-apoptotic protein Bcl-2.

Zhang et al. showed that UA dose-dependently decreased viability and induced apoptosis in both LNCaP androgen-dependent and LNCaP-A1 androgen-independent prostate cancer cells [39]. Western blot analysis revealed that UA treatment increased the phosphorylation/activation of c-Jun N-terminal kinase (JNK) and its downstream target transcription factor c-Jun without affecting the activation of other mitogen activated proteins kinases (MAPKs) such as the extracellular signal-regulated protein kinase (ERK) and p38. Fluorometric assays demonstrated an increase in caspase-3 and caspase-9 activity with no change in caspase-8 activity indicated activation of the intrinsic apoptotic pathway. UA treatment in both cell lines caused an increase in phosphorylation of Bcl-2 protein resulting in its degradation. The involvement of the JNK pathway in UA-induced apoptosis was confirmed using the JNK inhibitor, SP600125. The UA-induced activation of the examined apoptotic markers was abolished when cells were pretreated with SP600125. These data indicate apoptosis induction by UA is triggered by increased activation of JNK and downstream activation of the intrinsic apoptotic pathway [39].

LNCaP prostate cancer cells treated with UA (40 µM) had suppressed proliferation and increased apoptosis which occurred via mediation of the ROCK1/PTEN signaling pathway. UA induced cleavage of ROCK1 and phosphorylation of PTEN leading to increased protein expression of cytochrome c and cofilin-1. Increased cytochrome c due to UA treatment lead to increased activity of caspases-3 and -9 [40].

The treatment of primary malignant tumor (RC-58T/h/SA#4)-derived human prostate cancer cells with UA inhibited survival, reduced cell density, and increased apoptosis. Cell treatment with UA (30 or 40 µg/mL, 24 h) caused nuclear condensation, formation of apoptotic bodies, and DNA fragmentation, all markers of apoptosis. Further investigation of the cell cycle with flow cytometry showed an increased number of cells in the subG1 phase. An increase of active caspase-3, -8, and -9 was observed with UA treatment (investigated by fluorometric assays). Western blot analysis showed upregulation of the pro-apoptotic protein Bax, downregulation of the anti-apoptotic protein Bcl-2, and induction of Bid cleavage. These data suggest that UA treatment stimulates the activation of caspase-8 leading to Bid cleavage and downstream activation of caspase-9. UA also increased the expression of mitochondrial apoptosis factor (AIF) and caused its translocation into the nucleus. Together these data suggest that treatment of RC-58T/h/SA#4 prostate cancer cells with UA induces apoptosis through both caspase-dependent and independent pathways [41].

**Table 1 ijms-24-07414-t001:** Effects of ursolic acid against prostate cancer: summary of in vitro studies.

Cell Type	Dose	Findings	Mechanism	Reference
PC-3 LNCaP	UA 55 µM (PC-3) 45 µM (LNCaP)	↓ Cell survival ↑ Apoptosis	↓ Bcl-2 protein	[38]
LNCaP LNCaP-A1	UA 10, 20, 50, 80, 100 µM	↓ Cell viability ↑ Apoptosis	↑ p-JNK ↑ p-c-Jun ↑ caspase-3 activity ↑ caspase-9 activity ↑ p-Bcl-2 protein	[39]
LNCaP	UA 40 µM	↓ Cell proliferation ↑ Apoptosis	↑ cleaved ROCK1 ↑ p-PTEN protein ↑ cofilin-1 ↑ cytochrome c ↑ caspase-3 activity ↑ caspase-9 activity	[40]
RC-58T/h/SA#4	UA 40 µM	↓ Cell survival ↓ Cell density ↑ Apoptosis ↑ SubG1 cell population	↑ DNA fragmentation ↑ caspase-3,-8, and -9 activity ↑ PARP cleavage ↑ Bax protein ↓ Bcl-2 protein ↓ Bid protein ↑ AIF protein ↑ AIF nuclear translocation	[41]
PC-3	UA 80 µM	↓ Cell viability ↑ Apoptosis	↑ caspase-3, -8, and -9 activity ↑ caspase-8 and -9 cleavage ↑ p-JNK protein ↓ total-Bcl-2 protein ↑ p-Bcl-2 protein ↑ FasL mRNA and protein ↓ p-Akt ↓ MMP-9 levels	[42]
PC-3	UA 40 µM	↓ Cell viability G1 phase arrest	↑ PARP cleavage ↓ cyclin D1 and D3 protein ↓ CDK4 protein ↑ p21 protein ↑ LC3-II protein ↓ p-Akt protein ↓ p-mTOR protein ↓ p-p70S6K ↓ p-4EBP1	[43]
PC-3 LNCaP	UA 0–80 µM	↓ Proliferation ↑ Apoptosis	↓ Bcl-2 ↓ Bcl-xl ↓ survivin ↓ PI3K ↓ p-Akt protein ↓ p-mTOR protein ↑ cleaved caspase-3	[44]
DU145 LNCaP	UA 50 µM	↓ Cell proliferation ↑ Apoptosis	↓ p-AKT protein ↓ p-IkBα protein ↓ p65 protein and nuclear translocation ↓ p-IKKα/β protein ↓ NK-kB DNA binding ↓ p-STAT3 protein ↓ p-Src protein ↓ p-JAK2 protein	[45]
DU145	UA 50 µM	↓ Cell viability ↑ Apoptosis	↑ p-JNK protein ↑ p-c-Jun protein ↑ caspase-3, -9 activity ↑ p-Bcl-2, ↓ Bcl-2 protein	[46]
DU145	UA 25 µM	↑ Apoptosis	↑ ATP in cytosol ↑ P2Y_2_ mRNA ↑ COX-2 protein ↑ DNA fragmentation ↑ p-p38 ↑ p-Src protein	[47]
DU145	UA 10–40 µM	↓ Cell viability ↑ Apoptosis	↑ caspase-3 activity ↑ caspase-9 activity ↑ cyt-cytochrome c ↓ Mit-cytochrome c ↓ ROCK protein expression ↑ PTEN protein expression ↓ Cofilin-1	[48]
PC-3 LNCaP DU145	UA 35 µM 47 µM 80 µM	↓ Cell viability ↑ Cytotoxicity ↑ Apoptosis	↑cleaved PARP ↑ cleaved caspase-9 ↑ cleaved caspase-3 ↓ Wnt5α/β protein ↑ p-GS3β protein ↓ β-catenin	[49]
PC-3 LNCaP DU145	UA (30 µM) UA + TRAIL	↓ Cell viability ↑ Apoptosis	↑ cleaved PARP ↑ cleaved caspase-9 ↑ cleaved caspase-3 ↑ CHOP	[50]
HMVP2 DU145 PC-3 C4-2B	UA 20 µM	↓ Cell viability ↑ ROS	↓ ATP bioluminescence	[51]
PC-3 DU145 LNCaP	UA 50 µM	↓ Cell viability ↓ Cell Migration	↓ CXCR4 protein ↓ CXCR4 mRNA ↓ CXCL12	[52]
DU145	UA 2.2–21.9 µM		↓ MMP-2 activity ↓ MMP-9 activity	[53]
LNCaP PC-3	UA 20 µM	↓ Colony formation ↑ ROS		[54]
DU145 Exposed to radiation	UA 30 µM	↓ Cell survival ↑ Apoptosis ↑ DNA fragmentation	↑ cleaved PARP ↓ Bid	[55]

↑ = Increased; ↓ = Decreased.

Exposure of PC-3 prostate cancer cells to UA resulted in reduced viability and increased apoptosis through activation of both intrinsic and extrinsic apoptotic pathways. Fluorometric assays showed increased caspase-3, -8, and -9 activity while Western blotting demonstrated increased cleavage of caspase-8 and -9. Inhibiting either caspase-8 or -9 (using Z-IETD-FMK and Z-LEHD-FMK, specific inhibitors for caspase-8 and -9, respectively) prevented the UA-induced apoptosis. In addition, UA treatment lead to activation of JNK and subsequent Bcl-2 phosphorylation. UA reduced Akt phosphorylation and increased the levels of Fas ligand (FasL). FasL knockdown by a small interfering RNA (siRNA) approach attenuated the UA-induced caspase-8 activation and apoptosis. These data suggest FasL involvement in the UA-induced apoptosis of PC-3 prostate cancer cells. Cell invasion was also inhibited as evidenced by the downregulation of the matrix metalloproteinase-9 (MMP-9). Taken together, the data of this study support UA as a potential therapeutic agent against prostate cancer due to its ability to induce apoptosis via both the intrinsic and extrinsic pathways and to inhibit invasion and metastasis [42].

Shin et al. demonstrated that PC-3 prostate cancer cells treated with UA had reduced viability associated with cell cycle arrest at the G1 phase [43]. In addition, UA treatment enhanced the expression of the autophagosome marker LC3-II, clearly indicating induction of autophagy. These effects on autophagy occurred through the Beclin-1 and Akt/mTOR signaling pathways. Inhibition of autophagy with 3-methyladenine, and siRNA silencing of Belclin-1 and Atg5 lead to enhanced UA-induced apoptosis. These data suggest that in PC-3 cells, autophagy is a survival mechanism against UA-induced apoptosis and the use of autophagy inhibitors in combination with UA resulted in greater cancer cell inhibition. These data indicate that a combination of UA and autophagy inhibitors may provide a novel cancer therapy [43].

Meng et al. showed that prostate cancer cells LNCaP and PC-3 treated with UA had reduced proliferation and increased apoptosis [44]. UA caused a decrease in protein expression of Bcl-2, Bcl-xl, and survivin and an increase in activated caspase-3. Increased apoptosis in these cells was associated with reduced expression of phosphatidylinositol-3-kinase (PI3K) and reduced phosphorylation of the signaling proteins Akt and mTOR [44].

Androgen-independent DU145 and androgen-dependent LNCaP prostate cancer cells treated with UA had decreased proliferation and increased DNA fragmentation, an indicator of apoptosis [45]. NF-κB activity was suppressed through inhibiting TNF-α-induced IkB kinase (IKK) activation as well as IkBα and p65 phosphorylation. UA treatment of these cells also resulted in the suppression of STAT3 activation associated with suppression of the upstream kinases Src and JAK2. In these cell lines, UA treatment lead to the downregulation of NF-κB and STAT3 gene products. Importantly, this study was the first to demonstrate UA’s ability to suppress NF-κB activation in DU145 cells and TNF-α-induced NF-κB activation in LNCaP cells [45].

In a study by Zhang et al., human DU145 androgen-refractory prostate cancer cells were treated with UA resulting in a concentration-dependent decrease in cell viability. In addition, UA induced apoptosis as evidenced by fluorescence microscopy showing nuclear shrinkage, condensation, and fragmentation, all morphological changes typical of apoptotic cells [46]. In these prostate cancer cells, UA increased the phosphorylation of JNK indicating increased activation, with no effects on ERK1/2 or p38 MAPK. UA also increased the phosphorylation of the JNK activated transcription factor c-Jun. Pre-treatment of the cells with the JNK inhibitor SP600125 (10 µM, 2 h) abolished the UA-induced modulation of p-c-Jun, caspase-3, and p-Bcl-2, indicating that UA induces apoptosis through JNK activation in these cells [46].

DU145 prostate cancer cells had an increase in intracellular ATP and P2Y_2_ transcript levels when treated with UA (25 µM) [47]. Activation of P2Y_2_ led to activation of Src and phosphorylation of p38 leading to downstream overexpression of COX-2; COX-2 overexpression caused the cells to become resistant to apoptosis [47]. COX-2 overexpression was attenuated with suramin, a broad-spectrum P2Y inhibitor, added to the UA treatment; suramin added to the UA increased apoptosis, suggesting that P2Y is involved in UA resistance. Together these results suggest that initiation of apoptosis in DU145 prostate cancer cells is dependent on protein kinase C (PKC) activation after UA treatment. This study showed evidence that P2Y_2_ activation and subsequent COX-2 overexpression led to UA-induced apoptosis resistance. An important finding in this paper was the dual role of UA as a cancer treatment and its ability to both induce apoptosis and create resistance to it. Having a better understanding of the mechanisms involved in apoptosis resistance is important in designing cancer drugs [47].

Ursolic acid-induced apoptosis of DU145 prostate cancer cells is based on the increased activity of the pro-apoptotic proteins caspase-3 and caspase-9. Protein levels of cytochrome c were increased in the cytoplasm and suppressed in the mitochondria suggesting mitochondrial apoptosis pathway activation. UA in these cells suppressed the rho-associated protein kinase/phosphatase and tensin homolog (ROCK/PTEN) signaling pathway which led to the inhibition of cofilin-1 protein expression. Together, these data suggest that, in DU145 prostate cancer cells, UA induces apoptosis through ROCK/PTEN-mediated mitochondrial translocation of cofilin-1 [48].

Park et al. also found that treatment of PC-3, LNCaP, and DU145 prostate cancer cells with UA resulted in a dose-dependent increase in apoptosis [49]. In addition, UA treatment of these prostate cancer cells suppressed Wnt5α/β and β-catenin expression and increased phosphorylation of glycogen synthase kinase 3 β (GSK3β). Inhibiting GSK3β (SB216763) or using Wnt3a-conditioned medium resulted in reversal of the activation of apoptosis markers (cleaved caspase-3 and PARP) that was observed with UA treatment [49].

Ursolic acid treatment sensitized tumor necrosis factor-related apoptosis-inducing ligand (TRAIL)-resistant prostate cancer cells (LNCaP and DU145) allowing TRAIL-induced apoptosis to occur [50]. UA + TRAIL treatment caused a significant increase in caspase-3 activity and increased protein expression of cleaved PARP and cleaved caspase-9. UA treatment led to the upregulation of DR5 via CHOP. The overall results identified the use of UA as a sensitizer for TRAIL-induced apoptosis suggesting its potential as a combination treatment against prostate cancer [50].

Ventral prostate tumor cells derived from mice, HMVP2, were treated with 20 µM UA and a series of other natural compounds, including curcumin (CUR) and resveratrol (RES), to determine the most effective combinations [51]. This screening procedure was also performed with human prostate cancer cell lines DU145, PC-3, and C42B. Combination of UA with CUR resulted in the greatest suppression of cell viability. In addition, UA combined with either CUR or RES showed comparable suppression of cell growth/survival to the clinical agents docetaxel and enzalutamide. Treatment with UA alone or in combination with CUR or RES caused changes in the intracellular glutamine flux, suggesting changes to the citric acid cycle and therefore metabolic changes. Western blot analysis of the HMVP2 cells showed that UA (20 µM) caused changes to signaling molecules relevant to glutamate transport and apoptosis; UA decreased protein expression of ASCT2, p-Src (Tyr416), and p-STAT (Ser705). UA + CUR caused increased protein expression of p-AMPK; however, neither compound alone had any effect. UA caused decreased expression of p-S6 Ribo (Ser235/236) alone and with CUR while p-S6 Ribo (Ser240/244) was only decreased when treated with both UA + CUR. Both UA alone and with CUR lead to a significantly increased number of apoptotic cells (measured by Annexin V) and increased levels of cleaved PARP protein [51].

Shanmugam et al. showed that human prostate cancer cells (DU145, LNCaP, and PC-3) treated with UA not only had reduced viability, as measured by MTT assay, but also reduced cell migration [52]. In addition, treatment with UA resulted in dose-dependent downregulation of CXCR4 expression. This downregulation was found to be due to the downregulation of mRNA levels and was associated with reduced levels of NF-κB activation. UA inhibited the binding of NF-κB to the CXCR4 promoter. In addition, treatment with UA suppressed the CXCL12-induced migration and invasion of prostate cancer cells. [52].

UA extracted from *Vaccinium macrocarpon* (cranberries) inhibited the growth of DU145 prostate cancer cells. UA treatment reduced the activity of matrix metalloproteinases MMP-2 and MMP-9 as examined by gelatin gel electrophoresis [53]. These data demonstrate the anti-migratory and antimetastatic properties of UA.

In experiments by Wang et al., prostate cancer cells were treated with UA (20 µM) or phenethyl isothiocyanate (PEITC, 5 µM) and analyzed using qPCR and Western blotting. UA treatment resulted in changes to the mRNA levels of Setd7, Nrf2, quinone oxidoreductase 1 (Nqo1), and glutathione S-transferase theta 2 (Gstt2). In LNCaP cells, but not PC-3, UA treatment increased the protein expression of Setd7. Setd7 and Nqo1 protein expression were increased with PEITC (5 µM, 24 h). Short hairpin-RNA (shRNA) was used to knockdown Setd7 leading to the inhibition of colony formation (LNCaP and PC-3) and increased ROS (LNCaP cells). Chromatin immunoprecipitation assays showed that knockdown of Setd7 decreased H3K4me1 at the Nrf2 and Gstt2 promoter region and this effect was attenuated with either UA or PEITC treatments [54]. These data suggest that UA is able to induce Setd7 expression, activating the Nrf2/antioxidant response element signaling pathway, which protects DNA from damage due to oxidative stress.

DU145 prostate cancer cells treated with either UA (30 µM) or exposed to 5 Gy of irradiation had decreased survival. However, when irradiation and UA were combined, a greater decrease in cell viability and induction of apoptosis was seen. The increased apoptosis was associated with the activation of caspase-3 [55].

Overall, the articles presented here provide strong evidence for the effects of ursolic acid against proliferation and survival and induction of apoptosis of prostate cancer cells. In addition, the treatment of prostate cancer cells with UA resulted in inhibition of JNK, Akt, mTOR, p70 S6K, 4EBP1, and NF-κB (Figure 2). Moreover, cleaved caspases and PARP, indicators of apoptosis, were increased as well as the tumor suppressor PTEN. In addition, UA inhibited MMP-2 and -9 indicating antimetastatic effects.

### 2.2. Effects of Ursolic Acid against Prostate Cancer: Evidence from In Vivo Studies

Only a few studies have utilized mouse xenograft models of prostate cancer to examine the effects of ursolic acid in vivo (Figure 3). HMVP2 prostate cancer cells, grown as spheroids, were subcutaneously injected into the flanks of immunocompromised mice and allowed to grow for 13 days. UA and the polyphenols resveratrol and curcumin alone or in combination were then added to the diet of the animals and the treatment continued for 32 days. Tumor volume was monitored starting on day one and throughout the experimental time. Administration of UA into the diet had no effect on animal body weight or daily food consumption. Treatment with UA alone caused small reductions in tumor volume and weight. However, when UA was combined with curcumin or resveratrol, a greater reduction was seen, with the combination of UA and curcumin showing the greatest effect [51] (Table 2). Unfortunately, whether such a combination treatment affects the bioavailability of either of the chemicals used is not known (was not examined) and hopefully it will be addressed in future studies.

In another study, immunodeficient mice transplanted with human VCaP-Luc prostate cancer cells had reduced tumor growth when administered ursolic acid. Cellular metabolites and metabolism-related signaling pathways were regulated by UA including S-adenosylmethionine (SAM), suggesting potential methylation reprogramming resulting in overall anticancer effects [56].

Shanmugam et al. fed transgenic adenocarcinoma of mouse prostate (TRAMP) mice a diet containing 1% *w/w* UA continuously for 36 weeks, which initially resulted in delayed formation of prostate intraepithelial neoplasia (PIN). During weeks 12–18 of the experiment, there was an inhibited progression of the PIN to adenocarcinoma. UA treatment reduced tumor growth and prolonged overall survival without affecting body weight. Subsequent experiments revealed that UA treatment led to downregulated activity of pro-inflammatory mediators such as NF-κB, STAT3, AKT, and IKKα/β in prostate tissues and reduced serum levels of TNF-α and IL-6. Immunohistochemical analysis of tumor tissue samples revealed reduced expression of cyclin D1 and COX-2 and increased levels of caspase-3. The serum samples had nanogram levels of detectable UA. Taken together these data support the use of UA for both prevention and treatment of prostate cancer [57].

Nude mice were injected subcutaneously with DU145 prostate cancer cells and then fed a diet containing 200 mg/kg UA twice a week for 6 weeks [45]. Mice fed UA had a lower tumor volume than the vehicle control group (DMSO), while overall body weight remained the same. Immunohistochemical analysis of the removed tumor tissues showed a decrease in expression of VEGF and increased expression of caspase-3 [45].

Transgenic adenocarcinoma of mouse prostate (TRAMP) mice treated with UA showed suppressed CXCR4 expression in the prostate and inhibited metastasis of prostate cancer to distal organs including the liver and lungs. CXCR4 is an important signaling molecule in cancer metastasis and its downregulation with this treatment suggests UA has potential as a treatment against prostate cancer cell metastasis [52].

A transgenic line of mice was created in which the PTEN tumor suppressor gene was specifically knocked out in prostate tissues (PTEN KO). PTEN KO mice had an age dependent increase in the size of the prostate lobes, and this increase was attenuated in mice fed a diet containing UA. Epigenomic CpG methyl-seq analysis showed that UA was able to attenuate the differentially methylated regions induced in PTEN KO mice [58].

**Table 2 ijms-24-07414-t002:** Effects of ursolic acid against prostate cancer: summary of in vivo studies.

Model	Dose/Duration	Findings	Mechanism	Reference
Allograft mouse HMVP2 cells	UA UA + CUR UA + RES	↓ Tumor volume ↓ Tumor weight	Not investigated	[51]
Male NCr immunodeficient mice VcaP cells injected subcutaneously	UA 0.1% (*w/w*) Orally 8 weeks	↓ Tumor growth	↑ SAM	[56]
6-week nude mice DU145 cells injected subcutaneously	UA 200 mg/kg Orally Twice a week, 6 weeks	↓ Tumor volume	↓ VEGF ↑ caspase-3	[45]
TRAMP mice 4 weeks 12 weeks 24 weeks	UA 1% (*w/w*) 4–12 weeks 12–18 weeks 24–36 weeks	↓ PIN ↓ Tumor volume ↑ Overall survival	↓ p-STAT3 ↓ p-AKT ↓ p-IKKα/β ↓ serum TNF-α ↓ serum IL-6	[57]
TRAMP mice	UA 1% *w/w* 12 weeks	↓ Metastasis	↓ CXCR4	[52]
Prostate-specific PTEN KO male mice	UA 0.1% (*w/w*) Orally 6 and 14 weeks	∆ Methylation ∆ Gene expression	↓ *Has3* mRNA ↓*Cfh* mRNA ↓ *Msx1* mRNA ↑*BDH2* ↓ *Ephas* ↓ *Isg15* ↓ *Nos2*	[58]
Female athymic nude mice	UA 20–40 mg/kg 3 weeks	↓ Tumor volume ↓ Tumor weight	↓ p-Akt ↓ p-mTOR ↓ Ki67	[44]

↑ = Increased; ↓ = Decreased; ∆ = Changed.

Furthermore, UA abrogated the PTEN KO-induced prostate cancer-related oncogenes Has3, Cfh, and Msx1. Association analysis of these studies identified a correlation between methylation status and mRNA expression of the tumor suppressor gene BDH2 and oncogenes Ephas, Isg15, and Nos2. These data suggest that UA may regulate oncogenes and/or tumor suppressor genes through modulation of their promoter methylation at an early stage of tumorigenesis. A metabolomic study was also performed and found that UA attenuated the PTEN KO-induced cancer-associated metabolic changes. UA attenuated purine metabolism/metabolites as well as glycolysis/gluconeogenesis metabolism and pyruvate and lactate levels. These changes in metabolism suggest UA has an important role in PTEN KO-mediated metabolic and epigenetic reprogramming and therefore, UA protects against PTEN knockout-induced tumorigenesis [58]. This is an important finding as PTEN mutations drive many cancers including prostate cancer.

Female athymic nude mice treated with 20–40 mg/kg of UA intraperitoneally for 3 weeks resulted in a decrease in tumor volume and weight. Inhibition of proliferation and induction of apoptosis was evident. Immunohistochemical analysis revealed a decrease in p-Akt, p-mTOR, and Ki67 [44].

## 3. Effects of Ursolic Acid against Other Urogenital Cancers

### 3.1. Effects of Ursolic Acid against Renal and Bladder Cancer

A few studies also examined the effect of UA against renal and bladder cancers (Table 3). Experiments by Li et al. showed that UA decreased the viability of metastatic renal cell carcinoma (mRCC) 786-0 cells in vitro [59]. The UA parent compound induced G1 cell cycle arrest and apoptosis. UA exposure has also been associated with decreased invasiveness in A498 cells. Chen et al. reported that treatment of A498 renal cancer cells with UA decreased their proliferation and invasiveness compared to control untreated cells. The reduction in cell invasiveness was correlated with reduced MMP-2 levels. Moreover, the levels of NLR family pyrin domain-containing 3 (NLRP3) receptor, cleaved caspase-1, and IL-1β were significantly increased [60]. NLRP3 is associated with inflammasome formation and the use of an NLRP3 inhibitor (MCC950) abolished the UA-induced effects. These data indicate that UA activates the NLRP3 inflammasome pathway in A498 cells leading to downstream increased IL-1β expression and caspase-1 activation [60].

Gai et al. found that treatment of T24 bladder cancer cells with UA resulted in the inhibition of proliferation and induction of apoptosis. Treatment with UA resulted in inhibition of Akt and IκBα–NF-κB signaling, Bcl-2 downregulation, and caspase-3 upregulation [61]. These data showed that UA exerts its pro-apoptotic effect by suppressing Akt/NF-κB signaling in T24 cells (Table 3).

Non-toxic concentrations of UA have also been reported to pose antiangiogenic activity. Exposure to UA has been associated with decreased VEGF and iNOS expression levels in different tumor types [62]. UA also inhibits HIF-1α, which is involved in every step during angiogenesis [63]. It is well established that neo-angiogenesis plays a crucial role both in the development and progression of RCC. Antiangiogenic agents, together with immunotherapy, represent the cornerstone of contemporary treatment in patients with metastatic RCC [64]. It is therefore reasonable to assume that the addition of UA as a complementary agent together with the standard treatments, might provide an additional benefit to the patients. However, further research is needed with respect to the potential benefits and risks of UA administration in patients with metastatic RCC, before it is introduced into clinical practice.

### 3.2. Effects of Ursolic Acid Derivatives/Nanoformulations against Renal and Bladder Cancers

Ursolic acid derivatives have been suggested to pose antitumor activity against renal and bladder cancers (Table 4). A study conducted by Li et al. showed that the UA derivative FZU-03,010 decreased the viability of metastatic renal cell carcinoma (mRCC) 786-0 cells in vitro [59]. Treatment with FZU-03,010 induced G1 cell cycle arrest and apoptosis and inhibited the activation of signal transducer and activator of transcription 3 (STAT3). Induction of p21 and p27 cell cycle-dependent kinase inhibitor expression was observed. Finally, the investigators reported that the derivative promoted PARP and caspase-3/-7 cleavage [59].

Chadalapaka et al. reported that UA-derived analogs (2-position in the A-ring substitution) exhibited growth inhibition effects on KU7 and 253JB-V human bladder cancer cells [65] (Table 4). The 2-cyano and 2-trifluoromethyl derivatives were the compounds with the greatest anticancer activity. Similarly, Tu et al. exposed NTUB1 bladder cancer cells to twenty-three UA derivatives in vitro. Several compounds resulted in increased reactive oxygen species (ROS) formation that was associated with cell cycle arrest in G1 and G2/M phases, as well as tubulin polymerization inhibition and increased apoptosis [66].

Exposure of T24 human bladder cancer cells to UA resulted in a dose-dependent inhibition of growth and induction of apoptosis [67]. Interestingly, these effects were associated with a significant increase in the activation of the energy sensor AMPK. Knockdown of the catalytic (alpha) subunit of AMPK abolished the UA-induced effects while the use of the AMPK activator AICAR or transfection of the cells with a constitutive active form of AMPK mimicked the effects of UA. Moreover, the increased AMPK activation was associated with JNK activation, mTOR complex 1 (mTORC1) inhibition, and downregulation of survivin [67]. These data provide strong evidence of a significant role of AMPK in the UA-induced effects in T24 bladder cancer cells.

In another study by the same group, exposure of T24 cells to UA resulted in inhibition of clonogenic survival and induction of apoptosis that was associated with a significant increase in endoplasmic reticulum (ER) stress as seen by the increased phosphorylation of the ER membrane receptor double-stranded RNA-activated protein kinase (PKR)-like ER kinase (PERK) and increased levels of CHOP [68]. Treatment with UA increased JNK and apoptosis signal-regulated kinase 1 (ASK1) activation. Inhibition of ER stress by the use of a chemical inhibitor (salubrinal) or by silencing PERK significantly attenuated the UA-induced effects. The researchers also reported the induction of inositol-requiring enzyme 1 (IRE1)–tumor necrosis factor (TNF) receptor associated factor-2 (TRAF2)–ASK1 signaling complex formation, which resulted in the activation of ASK1–JNK pro-apoptotic signaling [68]. Overall, this study provides evidence of the role of ER stress in the UA-induced anticancer effects.

Lin et al. also reported that the antiproliferative and pro-apoptotic effects of a UA derivative (UA17) in NTUB1 human bladder carcinoma cells were associated with increased ROS and p53 levels, increased activation of p38 MAPK, and downregulation of Bcl-2 protein [69]. The combination of UA17 with cisplatin resulted in greater anticancer effects compared to each agent alone. Furthermore, daily administration (intratumorally) of UA17 in mice xenografted with MBT-2 murine bladder carcinoma cells resulted in significant reduction in tumor size and increased survival. The combination of UA17 with cisplatin was also effective in the murine MBT-2 bladder tumor model [69].

Huang et al. reported that UA enhanced gemcitabine-induced cytotoxicity in T24 and 5637 bladder cancer cells in vitro. Increased activation of JNK signaling pathway and inactivation of the PI3K/AKT signaling pathway was observed [70]. These data suggest that UA could be used as a complementary agent able to sensitize bladder cancer cells to clinically used chemotherapeutics.

Studies examining the ability of UA and its derivatives to sensitize tumor cells to chemotherapeutic agents as well as to ionizing radiation have been summarized in a review by Prasad et al. [71].

### 3.3. Effects of Ursolic Acid against Pheochromocytoma and Testicular Cancer

Jung et al. examined the anticancer effects of UA in preclinical models of pheochromocytoma [72]. The investigators reported that UA was cytotoxic to PC-12 cells and triggered apoptosis via reducing Bcl-2 levels, activating caspase-3, and inducing PARP cleavage. They also showed that UA induced the accumulation of p26 in PC-12 cells and promoted the conversion of microtubule-associated protein light chain 3 (LC3)-I to LC3-II [72]. These results suggest that UA induces autophagy in vitro, but impairs autophagy progression by blocking the downstream signaling pathway. Yoon et al. investigated the effects of UA from *Corni Fructus* in PC-12 cells in vitro [73]. They reported that UA inhibited iNOS expression, blocked the nuclear translocation of p65 subunit of NF-κB, and comprehensively inhibited NF-κB activity. UA treatment also reduced p-38, ERK1/2, and JNK phosphorylation [73]. On the other hand, Tsai et al. found that UA poses significant anti-oxidative and anti-inflammatory effects in PC-12 cells in vitro [74].

Although direct evidence from preclinical studies on the effect of UA and its derivatives on human testicular cancer models is lacking, indirect evidence suggests that it might provide some benefit as a complementary treatment. Both seminomatous and nonseminomatous testicular tumors are extremely chemosensitive as well as radiosensitive [75]. As previously mentioned, UA and its derivatives can enhance the in vitro antineoplastic effects of cisplatin and other agents, such as gemcitabine, in several models derived from different tumor types [69,70,71]. However, further research is needed before UA becomes a recommended therapy in patients with testicular cancer.

## 4. Bioavailability and Pharmacokinetics of Ursolic Acid

In the Biopharmaceutics Classification System (BCS), UA is classified as a type IV compound, being almost insoluble in water and exhibiting poor oral bioavailability and intestinal permeability [76]. UA is absorbed by the intestinal tract mainly through passive diffusion while the active transport of UA has also been reported, since UA probably acts as a substrate of P-gp permeability glycoprotein 1 (P-gp). UA is rapidly metabolized by the liver [77], contributing to its low bioavailability.

The pharmacokinetics of UA have been reported both in rodents and humans. In particular, plasma concentrations and tissue distributions of UA were measured by Chen et al. using liquid chromatography-mass spectrometry (LC-MS) in Sprague-Dawley rats. An oral dose of UA (10 mg/kg) was administrated, and plasma levels peaked at 1.1 µg/mL (2.4 µM) approximately 30 min after exposure while the highest concentration of UA was found in the lung [78]. Rats fed with Lu-Ying extract (80 mg UA/kg) by Liao et al. showed a plasma concentration of 294.8 ng/mL (645.5 nM) UA only 1 h after administration with a half-life of 4.3 h, implying either that UA has high binding activity in organs or is poorly absorbed by the intestine and metabolized by liver [79]. C57BL6 mice fed a diet supplemented with 0.5% UA reached a plasma concentration of 0.58 μg/mL (1.27 µM) after 8 weeks, while levels of UA in liver, colon, kidney, heart, bladder, and brain were increased through weeks 4 to 8, with the highest abundance (9.7 μg/g) observed in the liver [80].

In humans, Hirsch et al. reported that a single oral dose of UA (100 mg, 500 mg, and 1000 mg) had very low bioavailability in healthy adult volunteers (14 subjects) [81]. Since the gastrointestinal mucosa permeability of UA is poor and its oral absorption rate is low, UA-related nanoformulations, including liposomes, polymeric nanoparticles, and polymeric micelles, are designed in order to be administered mainly intravenously and to improve the drug delivery efficiency to the site of action.

A number of pre-clinical or clinical studies has been conducted the last ten years to investigate the bioavailability and pharmacokinetics of UA-related formulations which indicated that liposomes are more effective and safer compared to nanospheres and polymeric micelles [82]. For example, Xia et al. enrolled eight healthy volunteers in a single-dose study of ursolic acid liposomes (98 mg/m^2^) and the maximum plasma concentration observed was 3404.6 ± 748.8 ng/mL [83]. Twenty-four healthy volunteers were enrolled in a single-dose study by Zhu et al. and were randomly assigned to 37, 74, and 98 mg/m^2^ doses, while eight patients were to a multiple-dose study and received 74 mg/m^2^ doses for 14 consecutive days. The results showed that the maximum plasma concentration increased linearly with dose escalation (1835, 2865, and 3457 ng/mL, respectively) while no drug accumulation was observed after repeated administration [84].

Wang et al. enrolled 63 subjects for a single-dose study of UA liposomes, which presented linear values of pharmacokinetic parameters and showed that the maximum tolerated dose of UA liposomes was 98 mg/m^2^ [85]. The same results were observed by Qian et al. who conducted a phase I clinical trial with 21 participants to evaluate the efficacy and tolerability of UA liposomes [86]. Overall, the pharmacokinetics profiles of UA liposomes are linear and proportional to the dosage in a range from 37 mg/m^2^ to 98 mg/m^2^, indicating that they are a more effective means for delivering UA.

The toxicity and possible side effects of UA administration in humans has not been extensively studied. In one study by Wang et al. [85], single doses of UA (11, 22, 37, 56, 74, 98, and 130 mg/m^2^) were administered by a 4 h intravenous infusion, to four patients and 35 healthy adult volunteers to evaluate toxicity. The dose-limiting toxicity (DLT) was observed at 74, 98, and 130 mg/m^2^, and included hepatotoxicity and diarrhea. Other reported adverse effects were mild and included nausea, abdominal distention, microscopic hematuria, elevated serum sodium, and skin rash [85]. Overall, no studies examining in detail the toxicity of oral administration of UA in humans exist and future studies are required to address this important issue.

## 5. Conclusions

The results from in vitro studies, reviewed herein, suggest that UA is active against prostate cancer cells inducing apoptosis and inhibiting cell cycle progression. The findings from a few animal studies indicate that UA could reduce tumor volume and increase survival in mice xenografted with prostate cancer cells. In addition, there are evidence from in vitro studies indicating the effects of UA against renal, bladder, and testicular cancers. A search of the literature did not reveal studies examining the combination of current chemotherapy agents such as docetaxel or paclitaxel and UA. Such studies will provide information about whether UA could act as a chemosensitizer. More animal studies as well as clinical trials should be performed in the future to determine the potential of UA to be used as an anti-cancer agent alone or in combination with other chemotherapy drugs to enhance their efficacy and potentially reduce the side effects experienced by patients. In addition, more animal and human clinical studies are required to examine the possible adverse effects and toxicity of oral administration of UA.

## Figures and Tables

**Figure 1 ijms-24-07414-f001:**
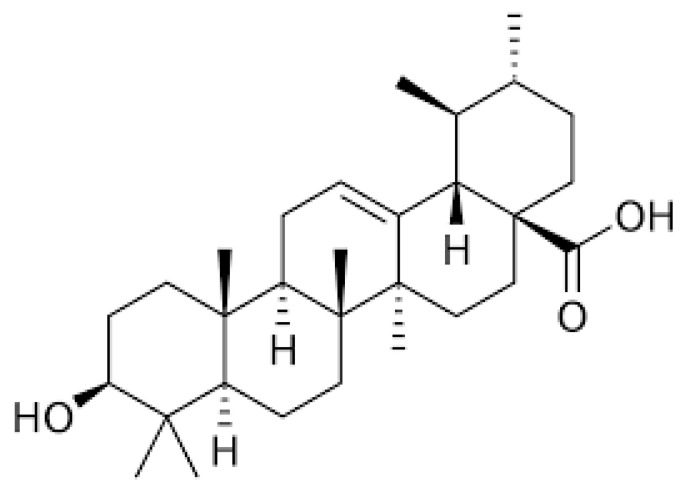
Chemical structure of ursolic acid (UA). Figure created in BioRender.com.

**Figure 2 ijms-24-07414-f002:**
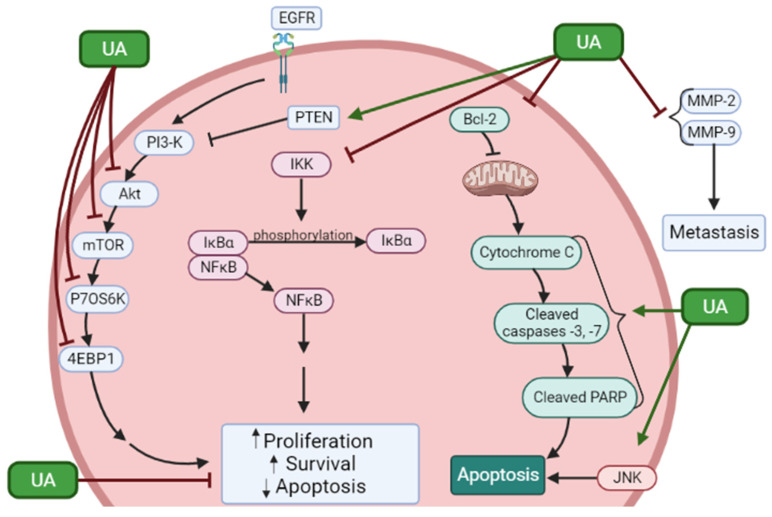
Effects of ursolic acid on signaling molecules in prostate cancer cells. UA reduced proliferation and survival and induced apoptosis of prostate cancer cells. These effects were associated with inhibition of phosphorylation/activation of JNK, Akt, mTOR, p70 S6K, and 4EBP1 and inhibition of the NF-κB pathway. Increased levels of PTEN and apoptosis markers cleaved caspases and PARP were seen. In addition, UA inhibited MMP-2 and -9 indicating antimetastatic effects. The figure was created using BioRender.com based on the data presented in Table 1. ↑ = Increased; ↓ = Decreased.

**Figure 3 ijms-24-07414-f003:**
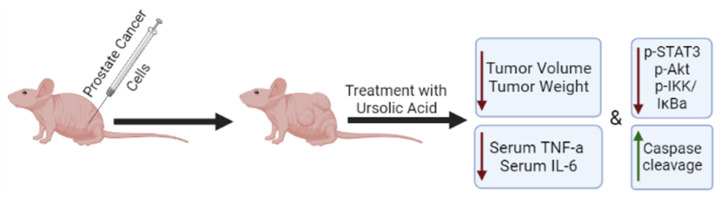
Effects of ursolic acid treatment on prostate cancer in vivo. Mice xenografted with prostate cancer cells and treated with ursolic acid, had reduced tumor volume and weight compared to untreated animals. The overall survival was also increased in UA-treated animals in one study. Ursolic acid treatment reduced the serum levels of TNF-α and IL-6 and the phosphorylation/activation of STAT3, AKT, and IKKα/β in tumor tissues. ↑ = Increased; ↓ = Decreased.

**Table 3 ijms-24-07414-t003:** Effects of ursolic acid against renal and bladder cancers: summary of existing evidence.

Cell Type	Dose/Duration	Findings	Mechanism	Reference
Renal Cancer				
786-O	UA 5 and 10 µM 48 h	↓ Cell viability ↑ Apoptosis	Cell cycle arrest G0/G1 phase	[59]
A498	UA 0.5 and 5 µM 12 h	↓ Cell viability ↓ Invasiveness	↑ NLRP3 ↑ caspase-1 ↑ IL-1β ↓ MMP-2	[60]
Bladder Cancer				
T24	UA 12.5, 25, 50 µmol/L 48 h	↓ Proliferation ↑ Apoptosis	↓ p-Akt protein ↓ p-IkBα protein ↓ NF-κBp65 protein and mRNA ↓ Bcl-2 protein and mRNA ↑ caspase-3 protein and mRNA	[61]

↑ = Increased; ↓ = Decreased.

**Table 4 ijms-24-07414-t004:** Effects of ursolic acid derivatives against renal and bladder cancers: summary of existing evidence.

Cell Type	Dose/Duration	Findings	Mechanism	Reference
Renal Cancer				
786-O	FZU-03,010 48 h	↓ Cell viability ↑ Apoptosis	Cell cycle arrest G0/G1 phase ↓ p-STAT3 ↑ p21 ↑ p27 ↑ Cleaved PARP ↑ Cleaved caspase-3 ↑ Cleaved caspase-7	[59]
Bladder Cancer				
KU7, 253JB-V	UA derivatives	↓ Proliferation	Not investigated	[65]
NTUB1	UA derivatives	↓ Proliferation ↑ Apoptosis	↑ G2/M phase ↑ G1 phase ↑ROS ↓ Tubulin polymerization	[66]
T24	UA derivatives	↓ Proliferation ↑ Apoptosis	↑ AMPK activation ↑ JNK activation ↓ mTORC1 activation ↓ Survivin	[67]
NTUB1	UA derivatives	↓ Proliferation ↑ G2/M phase ↑ G1 phase ↑ Apoptosis ↓ Tubulin polymerization	↑ROS	[68]
NTUB1	UA derivative (UA17)	↓ Proliferation ↑ Apoptosis	↑ROS ↑ p53 ↑ p38 MAPK activation	[69]
MBT-2 murine bladder cells injected to mice	UA derivative (UA17) 50, 100 mg/kg/day, intratumorally	↓ Tumor size ↑ Survival	↑ ROS	[69]
NTUB1	UA derivatives	↓ Proliferation ↑ G2/M phase	↑ ROS	[70]

↑ = Increased; ↓ = Decreased.

## Data Availability

Not applicable.

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
