# Peer review of "Ursolic Acid against Prostate and Urogenital Cancers: A Review of In Vitro and In Vivo Studies"

_ijms, 2023, doi:10.3390/ijms24087414_

Round 1

Reviewer 1 Report

The paper aims to summarize research studies examining the effects of ursolic acid and its derivatives against prostate and other urogenital cancers.

The text is very well written and structured. I recommend improving the conclusions and abstract of the study, highlighting the practical contributions of the research results.

Author Response

Reviewer 1:

English language and style are fine/minor spell check required

Comments:

The paper aims to summarize research studies examining the effects of ursolic acid and its derivatives against prostate and other urogenital cancers.

The text is very well written and structured. I recommend improving the conclusions and abstract of the study, highlighting the practical contributions of the research results.

Thank you for taking the time to review our manuscript. We followed your suggestions and improved the conclusions and abstract.

Reviewer 2 Report

In the present review, it was summarized research studies examining the effects of ursolic acid and its derivatives against prostate and other urogenital cancers.
The manuscript needs major and minor corrections.
Comments:
1. The English language of the manuscript should be thoroughly checked for grammar and writing.
2. Consider the section titled UA nanoformulations prepared to fight cancer.
3. The section "2.1. Effects of ursolic acid against prostate cancer: evidence from in vitro studies" is very long and it is better to write more briefly because most of the studies mentioned in the table are also mentioned.
4. Some contents are not referenced, for example, lines 70, 78, 209, and 330.
Lines 83 and 86 do not have references, and the following references are recommended:
PMID: 36804123
PMID: 33000538
PMID: 24518696
5. All et al., in vivo and in vitro should be written in italics.
6. Correct punctuation errors. For example, in line 47
7. The manuscript should be updated with new references (2022 and 2023).
8. Figure 3 is not mentioned in the text.
9. Line 19, is written twice.

Author Response

Reviewer 2:

Extensive editing of English language and style required

Please note that English is the first language of the first 2 authors.  The manuscript has been seen by all authors and attention paid to English language and style. The majority of the authors (university professors) are living in Canada (English speaking country) and have been communicating /teaching and publishing in English for 30-40 years.

Comments:

In the present review, it was summarized research studies examining the effects of ursolic acid and its derivatives against prostate and other urogenital cancers.

The manuscript needs major and minor corrections.

Comments:

  1. The English language of the manuscript should be thoroughly checked for grammar and writing.

Addressed/completed.

  1. Consider the section titled UA nanoformulations prepared to fight cancer.

Addressed/completed.

This comment was addressed. We have separated section 3 “Effects of ursolic acid against other urogenital cancers” into two subsections.

Section 3.1. “Effects of ursolic acid against renal and bladder cancer” includes studies focusing on the parent compound and the information is summarized in Table 3. Section 3.2. includes studies focusing on ursolic acid derivatives and is entitled “Effects of ursolic acid derivatives/nanoformulations against renal and bladder cancer”. This information is summarized in Table 4 within the review.

We believe these changes enhanced the quality of our manuscript.

  1. The section "2.1. Effects of ursolic acid against prostate cancer: evidence from in vitro studies" is very long and it is better to write more briefly because most of the studies mentioned in the table are also mentioned.

Addressed/completed.

Section 2.1. was edited to better present the studies in a succinct manner.

  1. Some contents are not referenced, for example, lines 70, 78, 209, and 330.

Lines 83 and 86 do not have references, and the following references are recommended:

PMID: 36804123

PMID: 33000538

PMID: 24518696

Addressed/completed. See the revised manuscript and below.

  1. All et al., in vivo and in vitro should be written in italics.

Addressed/completed. However, we leave it to the journal editorial team to determine if they should be italicized. 

  1. Correct punctuation errors. For example, in line 47

Addressed/completed.

  1. The manuscript should be updated with new references (2022 and 2023).

Addressed/completed.

A search of PubMed was conducted using the same key words as mentioned in this review. The search criterion was limited to the year 2022 to present and no additional studies (not already included) were observed.  

  1. Figure 3 is not mentioned in the text.

Addressed/completed.

  1. Line 19, is written twice.

Addressed/completed.

Reviewer 3 Report

Thank you for the interesting review article. I have these comments:

- In the abstract, the authors mentioned that Ursolic acid is a polyphenol. It is not correct. It is a triterpenoidal compound.

- The structure in Fig. 1 seems to be copied from other sources. Please draw it using Chem-Draw and then add in the manuscript.

- There are extensive information on the in vitro studies on UA and the compound seems to acts by multiple molecular mechanisms, however, if there are some literature data that can be merged together in one paragraph, I am asking the authors to do that. Also, one more point, what about the selectivity index (if so) reported in the literature showing the toxicity of UA on human cell lines. 

- The authors wrote << Many drugs used in chemotherapy are derived from plants. For example, the chemotherapy drugs paclitaxel and docetaxel, used in prostate cancer treatment, were originally isolated from the bark of the Pacific yew tree (Taxus brevifolia) and the needles of the European yew tree (Taxus baccata), respectively. Finding novel approaches to prevent and treat prostate and other urogenital cancers effectively is highly desirable and the search of plant-derived chemicals with strong anticancer potential is on-going.>> It is better to show the activity of paclitaxel and docetaxel, in prostate cancer treatment compared to Ursolic acid. 

- In the in vivo studies, there are few reports on the effectiveness of UA for prostate cancer management. However, It is better to show the toxicity of UA reported in the literature and the used dose in those previous studies.  

- The authors wrote <<Treatment with UA alone caused small reductions on tumor volume and weight. However, when UA was combined with curcumin or resveratrol a greater reduction was seen with the combination of UA with curcumin showing the greatest effect>> It is interesting point. Please highlight that point and mention in details the compounds which affects the bioavailability of UA. 

- In Table 3

- The style of references is written sometimes in brackets and sometimes et al., year.  

- In the same table, The authors showed UA derivatives have some activities. Please delete that and focus only on UA. It is not known whether the activity is due to UA or the substituents. 

Author Response

Reviewer 3:

English language and style are fine/minor spell check required

Comments:

Thank you for the interesting review article. I have these comments:

  1. In the abstract, the authors mentioned that Ursolic acid is a polyphenol. It is not correct. It is a triterpenoidal compound.

Addressed/completed. All subsequent mentions of the compound are addressed as such.

  1. The structure in Fig. 1 seems to be copied from other sources. Please draw it using Chem-Draw and then add in the manuscript.

Addressed/completed.

The figure was drawn using BioRender and is now cited.

  1. There are extensive information on the in vitro studies on UA and the compound seems to acts by multiple molecular mechanisms, however, if there are some literature data that can be merged together in one paragraph, I am asking the authors to do that. Also, one more point, what about the selectivity index (if so) reported in the literature showing the toxicity of UA on human cell lines.

Addressed/completed.

Text was altered to display information more succinctly where possible.

 We added more information regarding toxicity and adverse effects

of UA administration in humans.

The toxicity and possible side effects of UA administration in humans has not been extensively studied. In one study by Wang et al [85] single doses of UA (11, 22, 37, 56, 74, 98, and 130 mg/m2) were administered, by a 4-h intravenous infusion, to 4 patients and 35 healthy adult volunteers to evaluate toxicity.  The dose-limiting toxicity (DLT) was observed at 74, 98, and 130 mg/m2, and included hepatotoxicity and diarrhea. Other reported adverse effects were mild and included nausea, abdominal distention, microscopic hematuria, elevated serum sodium, and skin rash [85]. Overall, no studies exist examining in detail the toxicity of oral administration of UA in humans and future studies are required to address this important issue.

  1. The authors wrote << Many drugs used in chemotherapy are derived from plants. For example, the chemotherapy drugs paclitaxel and docetaxel, used in prostate cancer treatment, were originally isolated from the bark of the Pacific yew tree (Taxus brevifolia) and the needles of the European yew tree (Taxus baccata), respectively. Finding novel approaches to prevent and treat prostate and other urogenital cancers effectively is highly desirable and the search of plant-derived chemicals with strong anticancer potential is on-going.>> It is better to show the activity of paclitaxel and docetaxel, in prostate cancer treatment compared to Ursolic acid.

A search of PubMed was conducted:

  • Ursolic acid and docetaxel and prostate cancer
  • Ursolic acid and paclitaxel and prostate cancer

No results were found indicating that there no studies examining these combination treatments against prostate cancer.

  1. In the in vivo studies, there are few reports on the effectiveness of UA for prostate cancer management. However, it is better to show the toxicity of UA reported in the literature and the used dose in those previous studies.

  1. The authors wrote <<Treatment with UA alone caused small reductions on tumor volume and weight. However, when UA was combined with curcumin or resveratrol a greater reduction was seen with the combination of UA with curcumin showing the greatest effect>> It is interesting point. Please highlight that point and mention in details the compounds which affects the bioavailability of UA.

This study was evaluated closely, and bioavailability was not studied with respect to combination treatments. This gap in literature was noted and added within our review. Future studies should consider how bioavailability is affected/differs in combination treatments.

 The text in the revised manuscript has been changed as follows:

Treatment with UA alone caused small reductions on tumor volume and weight. However, when UA was combined with curcumin or resveratrol a greater reduction was seen with the combination of UA with curcumin showing the greatest effect [51] (Table 2). Unfortunately, whether such combination treatment affects the bioavailability of any of the chemicals used is not known (was not examined) and hopefully it will be addressed in future studies.

  1. In Table 3
    1. The style of references is written sometimes in brackets and sometimes et al., year.

Addressed/completed.

  1. In the same table, the authors showed UA derivatives have some activities. Please delete that and focus only on UA. It is not known whether the activity is due to UA or the substituents.

Addressed/completed. We feel that there is value in showing UA derivatives and for such reasons we have decided to keep this information. However, we have separated   section 3 “Effects of ursolic acid against other urogenital cancers” into two subsections.

Section 3.1. “Effects of ursolic acid against renal and bladder cancer” includes studies focusing on the parent compound and the information is summarized in Table 3.

Section 3.2. includes studies focusing on ursolic acid derivatives and is entitled “Effects of ursolic acid derivatives/nanoformulations against renal and bladder cancer”. This information is summarized in Table 4 within the review.

We believe these changes enhanced the quality of our manuscript.

Reviewer 4 Report

The review titled "Ursolic acid against prostate and urogenital cancers: a review of in vitro and in vivo research" examined the capabilities of urolic acid against prostate and other urogenital (or genitourinary) cancers. The evaluation is exhaustive and well-written. It will benefit the scientific community; hence, I suggest publication in its existing form.

Author Response

Reviewer 4:

English language and style are fine/minor spell check required

Comments:

The review titled "Ursolic acid against prostate and urogenital cancers: a review of in vitro and in vivo research" examined the capabilities of urolic acid against prostate and other urogenital (or genitourinary) cancers. The evaluation is exhaustive and well-written. It will benefit the scientific community; hence, I suggest publication in its existing form.

Thank you for taking the time to review our manuscript.

We have addressed other reviewer’s comments and the revisions have further enhanced the quality of our manuscript.

Round 2

Reviewer 2 Report

The manuscript has been corrected and is acceptable.